# A Protocol for the Isolation of Oval Cells without Preconditioning

**DOI:** 10.3390/ijms251910497

**Published:** 2024-09-29

**Authors:** Rocío Olivera-Salazar, Aránzazu Sánchez, Blanca Herrera, Juan García-Sáez, Luz Vega-Clemente, Pedro Villarejo Campos, Damián García-Olmo, Mariano García-Arranz

**Affiliations:** 1New Therapies Laboratory, Health Research Institute-Fundación Jiménez Díaz University Hospital (IIS-FJD), Avda. Reyes Católicos, 2, 28040 Madrid, Spain; rocio.olivera@quironsalud.es (R.O.-S.); luz.vega@quironsalud.es (L.V.-C.); damian.garcia@quironsalud.es (D.G.-O.); 2Department of Biochemistry and Molecular Biology, Faculty of Pharmacy, Complutense University of Madrid (UCM), Health Research Institute of the “Hospital Clínico San Carlos” (IdISSC), 28040 Madrid, Spain; munozas@ucm.es (A.S.); blancamh@ucm.es (B.H.); juagar14@ucm.es (J.G.-S.); 3Biomedical Research Networking Center in Hepatic and Digestive Diseases (CIBEREHD-ISCIII), 28029 Madrid, Spain; 4Department of Surgery, Fundación Jiménez Díaz University Hospital (FJD), 28040 Madrid, Spain; pedro.villarejo@quironsalud.es; 5Department of Surgery, Universidad Autónoma de Madrid, 28034 Madrid, Spain

**Keywords:** pretreatment-free cell isolation protocol, hepatic progenitor cells (HPCs), oval cells (OCs), hepatotoxic treatment, mice liver models, differentiation capacity, liver diseases

## Abstract

Oval cells (OCs) is the name of hepatic progenitor cells (HPCs) in rodents. They are a small population of cells in the liver with the remarkable ability to proliferate and regenerate hepatocytes and cholangiocytes in response to acute liver damage. Isolating OCs generally requires a pretreatment with special diets, chemicals, and/or surgery to induce hepatic damage and OC proliferation in mice. Unfortunately, these pretreatments are not only painful for the mice but also increase the cost of the assays, and the effects on the different organs as well as on various liver cells are still unclear. Therefore, the search for a protocol to obtain OCs without prior liver damage is mandatory. In our study, we present a protocol to isolate murine OCs from healthy liver (HL-OCs) and compare them with OCs isolated from mice pretreated with 3,5-diethoxycarbonyl-1,4-dihydrocollidine (DDC-OCs). Our results demonstrated that cells derived from untreated mice exhibited similar behavior to those from treated mice in terms of surface marker expression, proliferation, and differentiation capacity. Therefore, given the impracticability of isolating human cells with prior hepatotoxic treatment, our model holds promise for enabling the isolation of progenitor cells from human tissue in the future. This advancement could prove invaluable for translational medicine in the understanding and treatment of liver diseases.

## 1. Introduction

Liver diseases are major contributors to illness and death worldwide. Despite their prevalence and the resulting liver dysfunction and failure, current medical treatments primarily focus on supportive care rather than curative approaches, except in cases of liver transplantation [1,2]. While the primary mechanism for liver regeneration involves the division of mature hepatocytes, evidence supports the existence of a hepatic niche in the periportal area of the adult liver (canals of Hering), where a small population of hepatic cells reside under certain conditions. These cells, also known as hepatic progenitor cells (HPCs), hepatic stem cells, liver stem cells, liver progenitor cells, or oval cells (OCs) in rodents [3,4,5], form a cell-type compartment in the periportal region. HPCs are not usually isolated from the normal adult liver, due to their low percentage, but they increase in number under specific circumstances associated with liver injury [5,6]. Therefore, HPCs are adult liver progenitor cells capable of regenerating the liver, mainly when mature hepatocytes are unable to solve damage [1,5,7,8,9]. As progenitor cells, HPCs have the ability to differentiate into hepatocytes and cholangiocytes, therefore being bipotential cells [10]. Although several studies suggest that they may give rise to other lineages under specific conditions in vitro, drawing the line to define these cells is not an easy task, and even their origin remains uncertain [6,11]. Nevertheless, the classical theory supports that HPCs/OCs derive from quiescent stem cells located in the canal of Hering, which is a structure connecting bile canaliculi formed by hepatocyte with bile ducts lined by cholangiocytes in the portal triad [12,13,14]. The study of HPCs has drawn attention not only because of therapeutic potential but also for better understanding of tissue homeostasis and regeneration of the liver [6].

Therefore, identifying, isolating, expanding in vitro, and utilizing HPCs pose significant challenges in the fields of tissue engineering and regenerative medicine [15]. Numerous dietary and chemical injury models have been used to induce the expansion of HPCs/OCs and study their biology [16]. Over the years, several methods and protocols have been developed for isolating OCs from rats and mice. These methods often involve different liver injury protocols designed to activate and expand OCs in rodents, including the use of chemicals alone or in combination with partial hepatectomy [2]. The most extensively used ones in rats and mice are the choline-deficient diet coupled with ethionine administration in mice (CD+E diet) [17,18,19,20,21] and the diet supplemented with the porphyria-inducing agent 3,5-diethoxycarbonyl-1,4-dihydrocollidine (DDC supplemented diet), which is predominantly utilized in mice [20,22,23,24]. Another approach to induce the expansion of OCs in mice and facilitate their isolation is retrosine treatment followed by partial hepatectomy [6]. Similarly, oval cell-inducing injury models involve the use of 2-acetylaminofluorene combined with partial hepatectomy in rats and mice (more commonly applied in rats) [2,8], cocaine/phenobarbital in mice [25], injection with *N*-acetyl-para-aminophen in mice [26], and Dipin (1,4-bis [*N*,*N′*-di (ethylene) phosphamide] piperazine) injection in rats and mice, followed by partial hepatectomy or galactosamine in rats [27].

Given the imperative to isolate HPCs to explore their role in liver diseases, along with their potential for transplantation or treatment, this study aims to develop a novel protocol for isolating HPCs/OCs from healthy mice. We characterized these cells by comparing them with a pool of OCs extracted from mice fed with a DDC supplemented diet. The goal was to demonstrate that cells isolated using this new protocol exhibit behavior very similar to injured liver-derived OCs.

## 2. Results

In this study, we developed a new protocol for isolating hepatic progenitor cells without inducing liver damage in mice (HL-OCs). The average total number of oval cells extracted from mouse liver was around 1 million in HL-OC and 6, 5 million cells in pretreated mice fed with 3,5-diethoxycarbonyl-1,4-dihydrocollidine (DDC-OC) [28]. The extraction efficiency (number of oval cells/total number of cells extracted) for HL-OC was 3.9% while in DDC-OC it was between 10 and 15%, which is in agreement with the literature [29]. Cell viability in both protocols was around 70%. Once the OCs were isolated, we compared them with a pool of established oval cells (OCs) that had been isolated from what we will refer to as DDC-OCs.

The isolated cells have been confirmed to be oval cells by morphological analysis, CD markers, and their differentiation capacity, as shown in the following results.

### 2.1. Comparative Analysis of Cell Morphology

The morphology of HL-OCs was compared with DDC-OCs. Qualitative results showed that DDC-OCs were similar to HL-OCs when viewed at the same objective (Figure 1).

### 2.2. Comparative Analysis of Cell Proliferative Capacity

Cell proliferation was compared between DDC-OCs and HL-OCs at passages 5 and 10 (Figure 2). All cell types were able to proliferate for a week. At passage 10, from day 5, the cells reach the plateau, and the proliferation decreases in both cell groups. No differences were found between the two cell lines (DDC-OCs and HL-OCs).

### 2.3. Comparative Analysis of Surface and Hepatic Lineage Markers Expression

The expression profile of surface and hepatic markers was analyzed in DDC-OCs and HL-OCs at passages 5 and 10 by flow cytometry following incubation with the corresponding antibodies to compare the cell phenotype and lineage. The results revealed a significant decrease (*p*-values = 0.0008 *** and 0.0248 *, α = 0.05) in the CD34 surface marker expression at passage 10 in both cell lines (Figure 3). Additionally, only in DDC-OCs, a decrease in the CD90 marker expression was observed at passage 10 (*p*-value = 0.0790 **, α = 0.05) (Figure 3). Although, no significant alterations were observed in the other markers that were studied. No differences were found between the two cell lines at different passages.

### 2.4. Comparative Analysis of the Cell Differentiation Potential

As progenitor cells, OCs are able to differentiate into hepatocytes and share many characteristics with mesenchymal stem cells [27,30]. Therefore, for a more detailed characterization of these cells, it has been considered appropriate to differentiate them towards hepatocyte, cholangiocyte [31], adipocyte, and osteocyte lineages [32].

#### 2.4.1. Differentiation towards the Hepatocyte Lineage

All cell types were cultured with specific hepatic differentiation medium for 16 days. After this time, two complementary approaches were taken: cells were stained with PAS, which stains glycogen deposits characteristic of hepatocytes, and total RNA was isolated from the cells to analyze genes related to hepatic differentiation by RT-PCR. The results showed that both DDC-OCs and HL-OCs at passages 5 and 10 were capable of differentiating into the hepatocyte lineage (Figure 4 and Figure 5). No significant differences were found in the relative expression of hepatic differentiation genes (alpha fetoprotein, *Afp*; HNF1 homeobox A, *Hnf1a*; hepatic nuclear factor 4 alpha, *Hnf4α* and tyrosine aminotransferase, *Tat*) between passages and between DDC-OCs and HL-OCs (Figure 4).

#### 2.4.2. Differentiation towards the Cholangiocyte Lineage

DDC-OCs and HL-OCs were treated with cholangiocyte differentiation culture medium for one week, and then we analyzed the cholangiocyte differentiation by bright-field microscopy and by analyzing the expression of genes related to this process. After a week, the results showed that both DDC-OCs and HL-OCs at passages 5 and 10 were capable of differentiating into the cholangiocyte lineage (Figure 4 and Figure 5). We found a significant difference (*p*-value = 0.0299 *, α = 0.05) in HNF1 homeobox B (*Hnf1b*) gene expression and in HL-OCs along passages in culture, with higher expression in HL-OCs at passage 10 than at passage 5 (Figure 4). No significant differences were found in the expression of the remaining genes studied (sex-determining region Y box 9, *Sox*9; aquaporin 1, *Aqp1*; secreted phosphoprotein 1, *Spp*1). No significant differences were found between DDC-OCs and HL-OCs (Figure 4).

#### 2.4.3. Differentiation towards the Adipocyte Lineage

DDC-OCs and HL-OCs were cultivated under adipogenic culture conditions for 14 days, and then we analyzed the adipogenic differentiation by performing an oil red staining and by analyzing the expression of genes related to this process. Results showed that all cell types at passages 5 and 10 were able to accumulate lipids (Figure 4 and Figure 5). Furthermore, when adipogenic-related gene expression was studied, no significant differences were found between DDC-OCs and HL-OCs and between passages (Figure 4). No expression of leptin (*Lep*) and adiponectin (*Adipoq*) genes was found in any of the cell lines at any passages.

#### 2.4.4. Differentiation towards the Osteocyte Lineage

The DDC-OCs and HL-OCs were cultured under osteogenic culture conditions for 21 days, and then we analyzed the osteogenic differentiation by performing an Alizarin red staining and by analyzing the expression of genes related to this process. The results showed that both cell types independently of the passage were able to accumulate salts (Figure 5). However, none of the cells at any passage expressed the osteocalcin gene (bone gamma carboxyglutamate protein gene, *Bglap*) after culture with osteogenic differentiation medium (Figure 4). Significant differences were found in the expression of alkaline phosphatase gene (*Alpl*), which is only expressed in DDC-OCs, and expression levels significantly increased with passages (*p*-value = 0.0002 ***, α = 0.05). No significant differences were found in the expression of the remaining genes studied (Figure 4).

## 3. Discussion

The widespread use of liver transplantation is hindered by the limited availability of donor organs and is associated with significant risks [1,33]. Given that many liver disorders stem from hepatocyte dysfunction, there is increasing interest in transplanting isolated hepatocytes, contingent upon the availability of high-quality donor livers. While hepatocytes exhibit a robust capacity for in vivo replication, their isolation is technically challenging, and they cannot be efficiently expanded in vitro [1,16,33]. Conversely, hepatoblasts found in the embryonic liver are bipotent progenitors capable of generating both hepatic and biliary lineages, although their bipotential capability diminishes with development [6]. Hepatic progenitor cells (HPCs) from adult livers have garnered attention due to their therapeutic potential and utility in studying liver regeneration processes. Although the suitability of adult HPCs as a transplant substitute remains uncertain, understanding their regulation could potentially stimulate residual progenitor function in patients or guide the differentiation of induced pluripotent stem cells [1,2,4,6,16,34]. With liver transplantation currently the sole reliable treatment for extending the lives of patients with advanced liver diseases and donor liver scarcity posing a challenge, alternative therapeutic options are urgently needed [33]. The ability of HPCs to differentiate into parenchymal liver cells and their potential role in certain liver cancers underscore the importance of studying these cells for both therapeutic applications and their involvement in disease pathology [2,33,35].

For all these reasons, it is necessary to study the role of HPCs both for their therapeutic use and their involvement in various diseases and this is why a new protocol for isolating HPCs and oval cells in rodents (OCs) is necessary to enable their application in human livers.

Our method represents advancement in that it is less traumatic for the animals, as it does not induce prior liver damage. On the other hand, we do not know how this liver damage could affect other organs and specifically the viability of OCs. For example, chronic feeding of mice with 3,5-diethoxycarbonyl-1,4-dihydrocholidine (DDC) for 28 days caused cholestatic liver disease with a pronounced ductular reaction, hepatoperiductal fibrosis, and massive portal/periportal inflammatory infiltration [34]. It is important to note that the DDC diet causes accumulation of porphyrin intermediates, resulting in hepatobiliary injury [35], including segmental bile duct obstruction and pericholangitis involving extrahepatic bile ducts and altering bile acid metabolism [36]. Another study indicates that feeding mice a DDC diet for 28 days demonstrated an increase in hepatic gene expression of pro-inflammatory markers and profibrogenic markers in DDC-fed mice [37]. Regarding damage to other organs, excessive and persistent liver inflammation can lead to significant hepatocyte loss and worsen various liver conditions, such as ischemia–reperfusion injury, systemic metabolic disorders (e.g., obesity, diabetes, non-alcoholic fatty liver), alcoholic hepatitis, xenobiotic intoxication, and infections, ultimately resulting in irreversible liver damage, fibrosis, and carcinogenesis [38]. Chronic liver disease causes hepatocellular damage, which initiates a pro-inflammatory response in both parenchymal and non-parenchymal hepatic cells, eventually leading to liver fibrosis, cirrhosis, portal hypertension, and liver failure [39]. One should not forget that the liver is the largest gland in the human body, essential for lipid, glucose, and protein metabolism, and produces a variety of serum proteins. Its function is typically tested with proteins like aspartate aminotransferase (AST) and alanine aminotransferase (ALT), which can become elevated following liver stress, damage, or exposure to hepatotoxic drugs [40].

Due to all these alterations, there is a need to develop a protocol that does not cause liver damage, making its application in human liver biopsies potentially viable in the future. Here, we describe a new protocol for isolating OCs in healthy mice. Initially, we considered replicating the protocols already described for isolating these cells in our laboratory [28]. However, due to the inability to perform them in human livers, we considered isolating these cells with a simplified method. We have attempted this in our laboratory. Additionally, after liver extraction, we also aimed to use the most harmless reagents and materials possible for the cells. First, we did not perform perfusion or use different enzymes such as pronase E and DNase I used to isolate OCs [28]. We believe that this approach is simpler and more cost-effective. We also did not perform a Percoll gradient but carried out erythrocyte lysis instead, which replaced liver perfusion to remove erythrocytes. We did not use cloning rings to select clones, as we considered that after isolation, the non-adherent cells would be removed with the washes. By using a selective medium, we obtained a fairly homogeneous population of OCs from healthy mouse livers (HL-OCs) and were able to repeat this process three times. However, we did maintain the culture medium used for OCs isolated from mice fed with DDC (DDC-OCs), as we considered it to be the most suitable for simulating the hepatic microenvironment and for selecting the cells of interest. For all these reasons, we believe that the advantages of this protocol over conventional ones are, on the one hand, that it does not require prior administration of hepatotoxic agents, potentially enabling its use in human liver biopsies. On the other hand, by not administering hepatotoxic agents, it does not alter the viability and behavior of the isolated cells. Additionally, it is simpler and more cost-effective, which could allow more laboratories to use this protocol for research with these cells. As we have shown in the results, the cell population is quite homogeneous in all three cases, which contributes to their reproducibility.

Comparing the morphology and proliferation of DDC-OCs and HL-OCs, we observed that both cell types were similar. We found no significant differences in the expression of liver and surface markers between HL-OCs and DDC-OCs. We found that in both, CD34 expression was relatively low, probably because it is usually expressed in fetal liver and its expression varies and decreases in adult liver [41]. To confirm the potential of isolated cells, we differentiated the HL-OCs into hepatocyte, cholangiocyte, osteocyte, and adipocyte lineages and compared them with the DDC-OCs, showing that they exhibit very similar behavior to the DDC-OCs, except for the alkaline phosphatase (*Alpl*) gene expression in osteogenesis, which is also overexpressed in higher passages. High levels of ALPL have been associated with an increased risk of liver fibrosis [42] and bile duct obstruction [43], but further research is needed to draw more definitive conclusions.

On the other hand, we have observed that after differentiation to hepatocyte, the morphology found is not the typical polygonal hepatocyte morphology associated with histological sections; despite this, the hepatic differentiation genes were positive. One of the limitations of this study is that in hepatocyte differentiation, it would have been interesting to study functional markers such as bile salt export pump (BSEP) [44] and a cytochrome (for example, cyp2E1) [45] and to exclude cholangiocyte contamination by using markers such as CK19 [46]. As for cholangiocyte differentiation, we have shown this by imaging and cholangiogenesis-related gene expression (sex-determining region Y box 9, *Sox*9; aquaporin, *Aquap*; HNF1 homeobox B, *Hnf1b*; secreted phosphoprotein 1, *Spp*1). It would have been interesting to confirm the presence of cholangiocytes in the cultures by using markers such as CK19 or GGT staining [46] and to rule out the presence of hepatocytes using, for example, albumin [47].

Summarizing, we have not found significant differences in all the parameters analyzed between DDC-OCs and HL-OCs; therefore, we can conclude that this protocol may be a feasible and replicable alternative to isolate OCs in mice without the need for prior liver damage. We emphasize that their study is of mere importance given their implication in hepatic tumor development but also their potential therapeutic use in the field of regenerative medicine. Since this protocol has been validated in mouse liver, it will be applied to isolate HPCs from liver tissue biopsies, thereby enabling an in-depth study of the role of these cells in human liver diseases. All of this contributes to the potential for further research and application in human tissues in the future, ultimately aiming to treat and understand liver diseases.

## 4. Materials and Methods

### 4.1. Isolation and Culture of Oval Cells from Healthy Livers of Untreated Mice (HL-OCs)

All procedures described in this protocol were conducted in compliance with the guidelines established by the animal care committee at University Hospital Fundación Jiménez Díaz, following the standards outlined by the Spanish Council of Animal Care (Ref. PROEX 084/16).

For the isolation of liver cells, three C57BL/6J female mice aged between 4 and 12 weeks were euthanized. Prior to sacrifice, the mice were anesthetized using 2% isoflurane and 1% oxygen, followed by intracardiac injection of 200 µL of 2M potassium chloride for euthanasia. Following euthanasia, the abdomen of each mouse was shaved and disinfected with iodine solution. The intraperitoneal cavity was then accessed by carefully separating the layers using forceps to expose the liver. The liver was subsequently removed and placed in a 50 mL conical tube containing 15–20 mL of Hank’s Balanced Salt Solution (HBSS) (HyClone, South Logan, UT, USA) supplemented with 1% Zellshield (Minerva Biolabs, Berlin, Germany). The liver tissue was mechanically disaggregated in a Petri dish using scissors and two 1 mL pipette tips, and the resulting disaggregated tissue was collected by washing with HBSS in another 50 mL conical tube. The volume was adjusted to 10 mL with HBSS, and enzymatic digestion was carried out using 0.05% collagenase type I (Gibco, Grand Island, NY, USA) at 37 °C for 1 h with agitation.

After incubation, the collagenase activity was neutralized by adding 10 mL of fetal bovine serum (FBS) (Gibco, Paisley, UK) to the solution. In a second step, non-digested tissue was mechanically disaggregated using a 25G needle, and the resulting solution (containing both digested tissue and mechanically disaggregated tissue) was filtered through a 70 µm nylon cell strainer and washed with HBSS.

The filtered solution was centrifuged at 400× *g* for 5 min, and the resulting pellet was subjected to lysis of blood cells using a solution containing Tris 2M (VWR Prolabo Chemicals, Leuven, Belgium) and MgCl_2_ 1M (Alfa Aesar, Karlsruhe, Germany), in distilled water (dH_2_O), pH: 8, for 10 min with agitation, at room temperature (RT). After centrifugation, the supernatant was removed, and the pellet was resuspended in HBSS.

The isolated liver cells were seeded at a density of 2 × 10^5^ cells/cm^2^ on collagen I-coated (Corning, Bedford, MA, USA) dishes in Williams’ E medium (Gibco, Paisley, UK) supplemented with 10% FBS, 100 nM dexamethasone (Sigma Aldrich, St. Louis, MO, USA), insulin–transferrin–selenium–ethanolamine 1X (ITS) (Roche, Mannheim, Germany), 10 ng/mL mouse epithelial growth factor (mEGF) (Peprotech, Rocky Hill, NJ, USA), and 10 ng/mL mouse hepatic growth factor (mHGF) (R&D, Minneapolis, MN, USA). Zellshield 1% was added to the culture medium to prevent contamination, and the medium was changed weekly to remove non-adherent cells. After one month of culture, progenitor cells were replated weekly until passage 5 to 10 (Table 1).

### 4.2. Isolation and Culture of Oval Cells from DDC-Fed Mice (DDC-OCs)

To obtain DDC-OCs, a previously established protocol was used [28]. Nine-week-old C57BL/6J male wild-type (WT) mice were maintained on 0.1% 3,5-diethoxycarbonyl-1,4-dihydrocollidine (DDC-supplemented diet) for 13 days, and then the OC-enriched non-parenchymal cell fraction was isolated and plated. DDC-OCs were selected based on their characteristic epithelial morphology and subcultured for further expansion and characterization. DDC-OC line was harvested at 80–90% confluence using Tripsin-EDTA 1X (Gibco, Paisley, UK) and replated at 0.5–1 × 10^4^ cells/cm^2^ with two changes in medium per week until their use for the experiments. Two cell passages (passages 5 and 10) were used.

DDC-OCs were cultured with William’s E medium enriched with dexamethasone (100 nM), mEGF (10 ng/mL), mHGF (10 ng/mL) and ITS 1X. All cell cultures were maintained under 5% CO_2_ and 37 °C, and two changes in medium were performed per week (Table 1).

### 4.3. Morphology Comparison

To compare the morphology between DDC-OCs and HL-OCs at passages 5 and 10, the cells were seeded at 2 × 10^5^ cells/cm^2^ in 12-well plates. The pictures of the cultures were taken using an optical microscope Zeiss Axio Vert A.1 (Palex Medical, Madrid, Spain) and Image Software Zen 3.1.

### 4.4. Proliferation Assay

This assay was performed according to the manufacturer’s protocol for Deep Blue reagent (Invitrogen, Eugene, OR, USA) to compare the proliferative capacity between DDC-OCs and HL-OCs and between passages 5 and 10. The cells were seeded at 1.3 × 10^3^ cells/cm^2^ in 12-well culture plates. At 24 h, Deep Blue (10%) was added to the culture medium and incubated at 37 °C and 5% CO_2_ for 4 h. After incubation, 100 µL of medium of each sample was transferred to 96-well plates, and fluorescence was read (560 nm Excitation/590 nm Emission) on a TECAN EnSpire multimode Plate Reader (Perkin Elmer, Waltham, MA, USA) with Enspire Manager Software Version 4.

### 4.5. Flow Cytometry Analysis

For the characterization and comparison of DDC-OCs and HL-OCs at passages 5 and 10, flow cytometry was performed. The cells were harvested using trypsin EDTA 1X, and they were resuspended in 100 µL of cold Phosphate-Buffered Saline (PBS) at a density of 1 × 10^5^ cells per tube. The cells were incubated for 30 min at 4 °C, in dark conditions with the following mouse primary antibodies: CD11b, CD90, CD29, CD34, CD45, CD44, CD105, OV6, albumin, CK18, and CK19, according to each manufacturer’s instructions (Table 2). After immunostaining, cells were rinsed with PBS, and they were centrifuged at 200× *g* for 5 min. Then, the cells were acquired by Fast Canto II cytometer (Becton Dickinson, Franklin Lake, NJ, USA). The results were analyzed with FlowJo Software Version 10.

### 4.6. Differentiation Assays

Both DDC-OCs and HL-OCs at passages 5 and 10 were induced to differentiate into hepatocyte, cholangiocyte, adipocyte, and osteocyte lineages to assess their differentiation potential as progenitor cells and compare this across different passages. The induction of these differentiations was carried out with different specific culture media.

#### 4.6.1. Differentiation towards the Hepatocyte Lineage

To differentiate DDC-OCs and HL-OCs in the hepatocyte lineage, 1 × 10^5^ cells/well were seeded in a six-well plate. The following day, dimethyl sulfoxide 1% (DMSO) was added to culture medium, and it was maintained for four days. Then, the culture medium was removed, and fresh culture medium with 2.5 mM sodium butyrate (Sigma-Aldrich, St. Louis, MO, USA) was added. This condition was maintained for six days with medium replacement twice a week. Finally, the medium with sodium butyrate was removed, and fresh medium with 10 ng/mL mHGF was added. This new condition was maintained for six days. At the end of this experiment, the presence of hepatocytes was analyzed by Periodic acid–Shiff staining (PAS) (Sigma-Aldrich, St. Louis, MO, USA) according to the manufacturer’s instructions.

#### 4.6.2. Differentiation towards the Cholangiocyte Lineage

To differentiate the DDC-OCs and HL-OCs in the cholangiocyte lineage [48,49], 3 × 10^5^ cells/well were seeded in a six-well plate with DMEM/F12 (Gibco, Paisley, UK) (1:1) culture medium containing 20 ng/mL mHGF, 10% FBS, and 1% Zellshield, along with an equal volume of collagen I (Corning, Bedford, MA, USA) (pH 7) on ice. Then, the cell solution with collagen I was polymerized at 37 °C, 5% CO_2_ in a cell incubator. Once the solution was polymerized, 2 mL of the same DMEM/F12 (1:1) culture medium was added. The culture was maintained under these conditions for 7 days, adding 1 mL of fresh medium twice a week. After one week in culture, microscope pictures Zeiss Axio Vert A.1 (Palex Medical, Madrid, Spain) and Image Software Zen 3.1 were taken in bright field, and the cells were isolated from collagen I for RNA extraction using the following custom protocol: cell solution with collagen I and 2% collagenase type I (Gibco, Grand Island, NY, USA) were mixed in 15 mL tubes at a 1:1 ratio. This new solution was then incubated in a thermal block at 37 °C for 30 min at 500 rpm to dissolve the collagen type I. Once the collagen was dissolved, the cells were centrifuged at 200× *g* for 5 min to isolate the cell pellet. Finally, the cell pellet was frozen with 200 µL of PBS at −20 °C until RNA extraction and analysis of related cholangiocyte gene expression.

#### 4.6.3. Differentiation towards the Adipocyte Lineage

The DDC-OCs and HL-OCs were treated with differentiation Mouse Mesenchymal Stem Cell Functional Identification Kit (R&D, Minneapolis, MN, USA) with adipose supplement according to the manufacturer’s instructions for 14 days, with culture medium replacement twice a week. After 14 days, the cells were rinsed with PBS and fixed with 4% formaldehyde for 30 min. Then, the cells were rinsed again with PBS and fixed with 60% isopropanol for 5 min. The isopropanol was removed, and the lipid droplets accumulated were stained with Oil Red O (Acros Organics, Geel, Belgium) (3 mg/mL in 36% isopropanol) for 1 h at RT. Finally, the cells were washed with 60% isopropanol to remove excess stain and rinsed with PBS.

#### 4.6.4. Differentiation towards the Osteocyte Lineage

Mouse Mesenchymal Stem Cell Functional Identification Kit (R&D, Minneapolis, MN, USA) with osteogenic supplement was used according to the manufacturer’s instructions for 21 days with culture medium replacement twice a week. After 21 days, OC cultures were rinsed with dH_2_O and fixed with cold 70% ethanol for 1 h at RT. Then, the ethanol was removed, and cells were rinsed with dH_2_O. The calcium deposits were stained with Alizarin Red S (Sigma-Aldrich, St. Louis, MO, USA) (0.01 g/mL) for 1 h at RT, and then the cultures were rinsed with dH_2_O to remove the excess stain.

All differentiation experiments included their corresponding controls with conventional culture medium, and pictures were taken by optical microscopy Zeiss Axio Vert A.1 (Palex Medical, Madrid, Spain) and Image Software Zen 3.1. On the other hand, the differentiated cells were isolated and frozen with 200 µL of PBS at −20 °C until RNA extraction and analysis of related differentiation gene expression.

#### 4.6.5. Total RNA Extraction, cDNA Synthesis and Quantitative Reverse Transcription PCR (RT-qPCR)

The DDC-OCs and HL-OCs at passages 5 and 10 after each differentiation (towards hepatocyte, cholangiocyte, adipocyte, and osteocyte lineages) were frozen with 200 µL of 1X PBS at −20 °C until further analysis. Total cellular RNA was isolated using NZY total RNA isolation kit (NZYTech, Lisboa, Portugal). RNA yield and purity were analyzed using Nano Drop (Thermo Fisher Scientific, Eugene, OR, USA). Additionally, 1 μg of total RNA was used to generate cDNA using High-Capacity cDNA Reverse Transcription Kit (Thermo Fisher Scientific, Vilnius, Lithuania).

The samples were run in triplicate, and quantitative real-time PCR (qRT-PCR) was performed using SYBR Green (Roche, Basel, Switzerland), and amplified products were analyzed in ABI Prism 7900 HT Fast Real-Time (Applied Biosystems, Waltham, MA, USA). The relative expression gene method (2^−ΔCt^) and cycle threshold (Ct) were used as reference to analyze the results. Ct values were processed and normalized to reference gene (*Gusb*) in each sample. Primers used in this study for each differentiation are presented in supporting information.

### 4.7. Statistic Analysis

Statistical analyses were performed using GraphPad Prism version 6. All assays were conducted with three biological and technical replicates. Proliferation (Deep Blue) among different DDC-OCs and HL-OCs was compared using either the Mann–Whitney test or Student’s *t*-test. One-way ANOVA was used to compare cell surface markers assessed by flow cytometry. For evaluating changes in gene expression between early and late passages during differentiation and between cell types, Student’s *t*-test was applied. A *p*-value less than 0.05 was considered statistically significant, with α = 0.05.

## 5. Conclusions

Here, we have provided a suitable protocol for isolating oval cells from mice without exposure to hepatotoxic agents or surgery (HL-OCs). The cell population isolated and established in culture exhibits characteristics very similar to those of oval cells extracted from mice fed with a 3,5-diethoxycarbonyl-1,4-dihydrocollidine diet (DDC-OCs). Our protocol is simpler, cheaper, less invasive for the animals, and may have greater potential for research in human tissue for isolating human hepatic progenitor cells in the future.

## Figures and Tables

**Figure 1 ijms-25-10497-f001:**
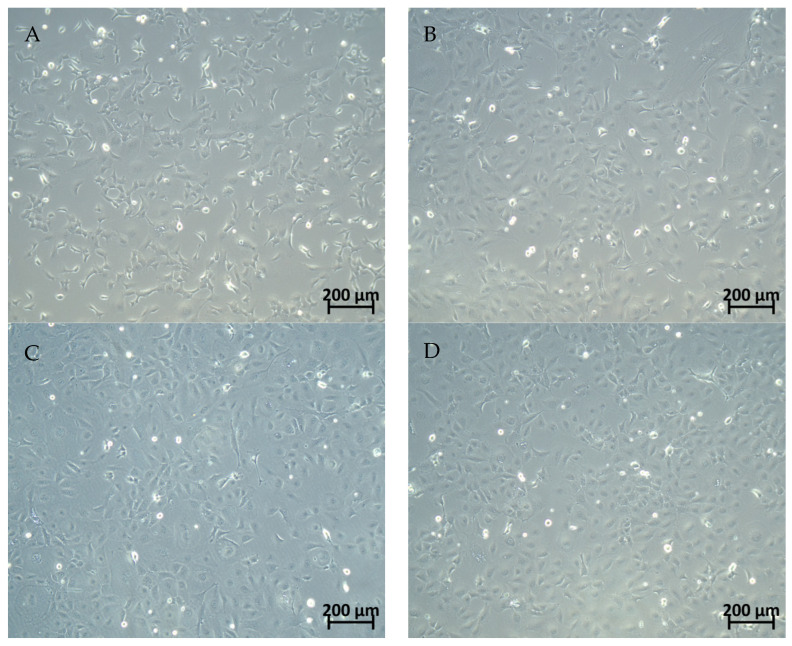
Morphological appearance in culture of pool of (**A**) oval cells isolated from mice fed with 3,5-diethoxycarbonyl-1,4-dihydrocollidine (DDC-OCs) or (**B**–**D**) oval cells isolated from healthy livers of untreated mice (HL-OCs). Three replicates per experiment. Bright field optical microscopy. Objective 10X.

**Figure 2 ijms-25-10497-f002:**
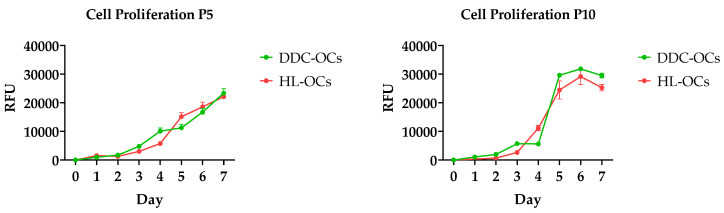
Cell proliferation of oval cells from mice fed with DDC diet (DDC-OCs) and oval cells from healthy liver of untreated mice (HL-OCs) at passages 5 and 10 was analyzed over 7 days in conventional culture medium. The experiments were performed in triplicate using the Deep Blue reagent according to the manufacturer’s instructions.

**Figure 3 ijms-25-10497-f003:**
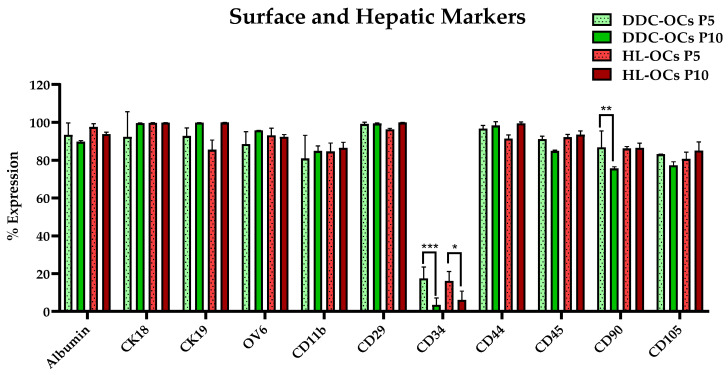
Mean with standard deviation of percentage expression of surface and hepatic epithelial lineage (albumin, CK18, CK19, and OV6) in oval cells isolated from mice fed with DDC diet (DDC-OCs) and oval cells isolated from healthy livers of untreated mice (HL-OCs) at passages 5 and 10. * = *p* < 0.05; ** = *p* < 0.01; *** = *p* < 0.001; α = 0.05, three replicates per experiment.

**Figure 4 ijms-25-10497-f004:**
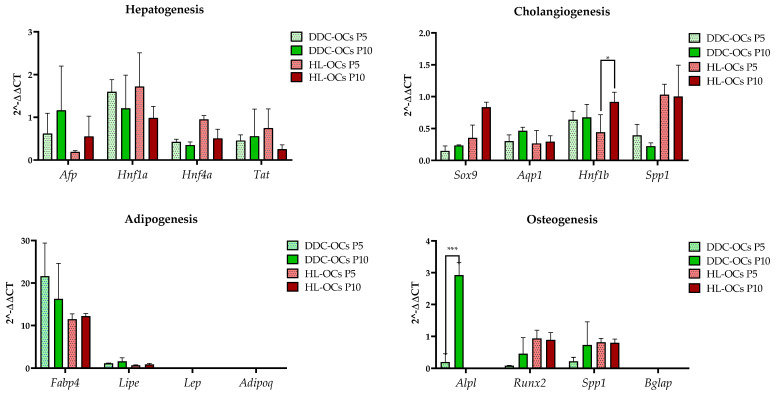
Differentiation multilineage-related relative mouse gene expression by RT-qPCR using Glucuronidase β (*Gusb*) as housekeeping gene (mean with standard deviation). Hepatic-related gene expression (alpha fetoprotein, *Afp*; HNF1 homeobox A, *Hnf1a*; hepatic nuclear factor 4 alpha, *Hnf4α*; tyrosine aminotransferase, *Tat*), adipogenic-related gene expression (fatty acid-binding protein, *Fabp4*; lipase, *Lipe*; leptin, *Lep*; adiponectin, *Adipoq*), cholangiogenesis-related gene expression (sex-determining region Y box 9, *Sox9*; aquaporin 1, *Aqp1*; HNF1 homeobox B, *Hnf1b*; secreted phosphoprotein 1, *Spp1*), and osteogenesis-related gene expression (alkaline phosphatase, *Alpl*; runt-related transcription factor 2, *Runx2*; secreted phosphoprotein 1, *Spp1*; bone gamma carboxyglutamate protein, *Bglap*) in oval cells isolated from treated mice fed with DDC diet (DDC-OCs) and oval cells isolated from healthy livers of untreated mice (HL-OCs) at passages 5 and 10. * = *p* < 0.05; *** = *p* < 0.001; α = 0.05, three replicates per experiment.

**Figure 5 ijms-25-10497-f005:**
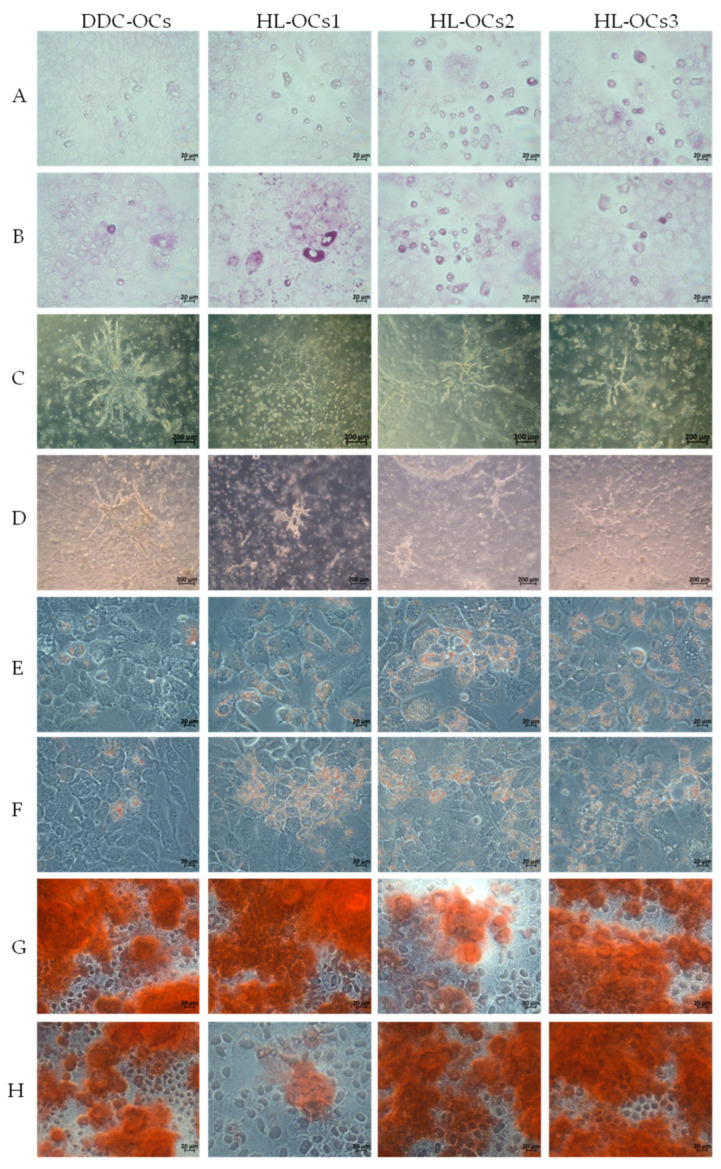
Multilineage differentiation of oval cells isolated from treated mice fed with DDC diet (DDC-OCs) and oval cells isolated from healthy livers of untreated mice (HL-OCs) into hepatogenesis (PAS staining, Objective 63X) using hepatocyte differentiation medium for 16 days of culture (**A**,**B**), cholangiogenesis using cholangiocyte differentiation medium for one week (bright field, Objective 10X) (**C**,**D**), adipogenesis using adipogenic differentiation medium for 14 days (oil red staining, Objective 63X) (**E**,**F**), and osteogenesis using osteogenic differentiation medium for 21 days of culture (Alizarin red staining, Objective 63X) (**G**,**H**) at passages 5 (**A**,**C**,**E**,**G**) and 10 (**B**,**D**,**F**,**H**). Three replicates per experiment.

**Table 1 ijms-25-10497-t001:** Comparison of the extraction of oval cells from healthy liver of untreated mice (HL-OCs) versus mice fed with a DDC diet (DDC-OCs).

Method	DDC-OCs		HL-OCs
Animals	C57BL/6J mice		C57BL/6J mice
Hepatotoxic treatment	DDC supplemented diet for 13 days.		Without treatment.
Anesthesia	Induction at 4% and maintenance at 2% isoflurane, 1% oxygen.		Induction at 4% and maintenance at 2% isoflurane, 1% oxygen.
Liver perfusion	Perfusion with Williams’ E medium with 0.1% *w*/*v* pronase E and 0.1% collagenase I.		Without liver perfusion.
Liver preservation	The extracted liver is placed in 50 mL conical tubes and washed with HBBS without Ca^2+^ and Mg_2_ (10 mL).		The extracted liver is placed in 50 mL conical tubes and washed with HBBS without Ca^2+^ and Mg_2_ (10 mL).
Mechanical disaggregation	The liver was triturated with scissors in Williams’ E medium with 0.1% *w*/*v* pronase E, 0.1% collagenase type 1 and 0.005% DNase I.		The liver was crushed with scissors and 2 × 1000 tips in a Petri dish in HBBS without Ca^2+^ and Mg^2+.^
Liver enzymatic digestion	The triturated liver was incubated in Williams’ E medium with 0.1% *w*/*v* pronase E, 0.1% collagenase type I, and 0.005% DNase I for 30 to 45 min at 37 °C in a shaking water bath.		The triturated liver was incubated in Williams’ E medium with 0.05% of collagenase I, 1 h, 37 °C in a shaking water bath.
Enzymatic inactivation	Inactivation of collagenase I with FBS medium at a final concentration of 1:2.		Inactivation of collagenase I with FBS medium at a final concentration of 1:2. Mechanical disaggregation of undigested tissue with 25 G needle.
Cellular filtering	Through 40 µm nylon mesh and cell sedimentation by centrifugation in culture medium.		Through 70 µm nylon mesh and cell sedimentation by centrifugation with HBSS.
Percoll gradient	Three consecutive washes with Williams’ E medium with 10% FBS, where the cells were purified with Percoll gradient (30% and 70% Percoll). The cell band is collected from the Percoll and washed with medium and FBS, and then the cells were centrifuged.		Without Percoll gradient.
Erythrocyte lysis	Without erythrocyte lysis.		Erythrocyte lysis with Tris solution, 2M MgCl_2_, 1M, in dH_2_O pH8, for 10 min under stirring.
Cell seeding	Cells were seeded at 2 × 10^5^ cells/cm^2^ on culture plates with Dulbecco’s modified Eagle’s/Ham’s F12 high-glucose medium with glutamine (1:1 mix) supplemented with 1 g/L insulin–transferrin–selenium, 1 g/L D-galactose, 0.3 g/L proline, 1.5 mmol/L, Na^+^ Pyruvate, 0.018M HEPES, penicillin/streptomycin, and 10% FBS.		Cells were seeded at 2 × 10^5^ cells/cm^2^ in collagen I pretreated plates with Williams’ E medium supplemented with 10% FBS and 100 nM dexamethasone, ITS 1X, 10 ng/mL mEGF, and 10 ng/mL mHGF.
Post cell seeding	Washing cells with PBS 1X and selection of single cells with cloning rings for further cloning and culture and expansion.		Cell washes with PBS 1X of non-adherent cells. No cloning rings.
Maintenance of cell culture	OC line was harvested at 80–90% confluence using Tripsin-EDTA 1X and replated at 0.5–1 × 10^4^ cells/cm^2^ in collagen I pretreated plates with Williams’ E medium supplemented with 10% FBS and 100 nM dexamethasone, 1× ITS, 10 ng/mL mEGF, and 10 ng/mL mHGF. Two changes in medium per week.		HPC line was harvested at 80–90% confluence using Tripsin-EDTA 1× and replated at 0.5–1 × 10^4^ cells/cm^2^ in collagen I pretreated plates with Williams’ E medium supplemented with 10% FBS and 100 nM dexamethasone, 1× ITS, 10 ng/mL mEGF, and 10 ng/mL mHGF. Two changes in medium per week.

**Table 2 ijms-25-10497-t002:** Mouse antibodies used for flow cytometry for characterization of oval cells isolated from DDC-fed mice (DDC-OCs) and oval cells isolated from healthy liver of untreated mice (HL-OCs).

Marker	Fluorochrome	Reference
CD11b	APC	Biolegend, San Diego, CA, USA
CD29	FITC	Invitrogen, San Diego, CA, USA
CD34	FITC	Invitrogen, San Diego, CA, USA
CD44	PE–Cy7	Invitrogen, San Diego, CA, USA
CD45	PE	Invitrogen, San Diego, CA, USA
CD90	FITC	Biolegend, San Diego, CA, USA
CD105	APC	Invitrogen, San Diego, CA, USA
OV6	PE	Santa Cruz, Dallas, TX, USA
CK18	FITC	Thermo Fisher Scientific, Regensburg, Germany
CK19	FITC	Thermo Fisher Scientific, Regensburg, Germany
Albumin	FITC	MyBioSource, San Diego, CA, USA

Allophycocyanin (APC); fluorescein isothiocyanate (FITC); R–phycoerythrin–cyanine 7 (Pe–Cy7); Phycoerythrin (PE).

## Data Availability

The data presented in this study are available upon request from the corresponding author due to (no data is available online).

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
