# Peer review of "A Protocol for the Isolation of Oval Cells without Preconditioning"

_ijms, 2024, doi:10.3390/ijms251910497_

Round 1

Reviewer 1 Report

Comments and Suggestions for Authors

The manuscript by Olivera-Salazar and colleagues aims to demonstrate the possibility of isolating oval cells (OCs), or hepatic progenitor cells, from rodent models without chemical, pharmacological or dietary preconditioning. This approach would allow, in the authors' idea, to isolate a good number of OCs while reducing ethical issues, since the animals would not be induced suffering due to treatments. The main results obtained are that the OCs obtained by the authors from untreated mice are morphologically, phenotypically and biologically similar to those obtained by inducing proliferation in vivo using DDC treatment. Furthermore, the authors are able to induce cellular differentiation of the obtained OCs towards hepatocyte lineages, cholangiocyte lineages and are able to induce adipogenic and osteogenetic differentiation. The work is potentially of great interest, mainly for obtaining OCs for studies on murine models, rather than for studies on human cultures, as correctly admitted by the authors. Some flows however diminish the enthusiasm for this manuscript:

- the most important issue concerns the quantity of OCs that can be isolated from untreated mice. In fact, the treatment induces a blooming of OCs that does not occur in normal mice, where they are an extremely minority fraction of the cells. In fact, the biliary structures are 3/5% of the liver cells and those present on the canals of Hering are a fraction of these. How many OCs can the Authors obtain from treated and untreated mice? Furthermore, what is the viability of the newly isolated OVs?

- the quality of the pictures in figure 1 is quite low, please replace them with better micrographs.

- there is some confusion in the text when you use the terms hepatic markers and hepatocyte markers, please clarify what you mean. In figure 3 I think I understand that with "hepatic" you mean hepatic epithelial lineage, phenotyping in fact includes stem cell, hepatocyte and cholangiocyte markers. In section 2.4.1 instead with hepatic differentiation you seem to mean differentiation towards the hepatocyte lineage, since you use mature hepatocyte markers. Please clarify these points well.

- Figure 4 should be implemented with some additional markers. In particular, for hepatogenesis, functional markers should be introduced, such as BSEP and a cytochrome (for example cyp2E1, or others) and not only phenotypic ones, in addition to a control that excludes the presence of cholangiocytes (for example cytokeratin 19). For cholangiocarcinogenesis, a purely biliary lineage marker (K19, for example) and alumina should be added, to verify the absence of hepatocyte contamination and the correct addressing towards the biliary phenotype.

- figure 5: please double check the figure legend, as the figure panels do not match those indicated in the legend. Also, the bright field photos of cholangiocytes are not sufficient to identify them as true biliary structures. I suggest histological staining for GGT for confirmation.

Author Response

Dear reviewer,

I hope this message finds you well. We would like to thank you for your time and attention to our manuscript titled "A Protocol for the Isolation of Oval Cells without Preconditioning."

We are pleased to inform you that we have completed all necessary revisions in response to your comments in the attached word.

We greatly appreciate the thoughtful feedback provided and are committed to addressing the suggestions in the best possible way.

We look forward to hearing back from you and will promptly respond to any further queries or requirements.

Thank you once again for your time and consideration.

Sincerely,
Mariano García Arranz and Rocío Olivera Salazar

Reviewer 2 Report

Comments and Suggestions for Authors

The isolation of HPCs to explore their role in liver diseases, along with their potential for transplantation or treatment is important. Established protocolls for the isolation of OCs generally requires the induction of hepatic damage with OC proliferation in mice. As the induction of liver damage may be painful for the mice, increases the cost, and may have unwanted effects on various liver cells, a protocol for isolation of OCs from healthy liver was established. The authors demonstrated that the OCs from healthy liver have same features in terms of surface marker expression, proliferation, and differentiation capacity compared OCs isolated from DDC treated mice. The experiments and results appear well suited to support the conclusions. The manuscript is well written and logically structured.

Some points could be improved:

1. Efficiency of both methods should be compared. The authors should present absolute numbers of OCs isolated from livers of healthy and DDC treated mice.

2. line 42: canal of hering

3. Figure 1: Oval cells have a diameter of ~10µm (https://doi.org/10.1073/pnas.1734199100). The authors should check the micrographs and size bars as cells appear much bigger, especially in B, C, and D. If this is an artefact of the cell culture condition, it should be mentioned and discussed appropriately.

4. Figure 1: staining and appearance of the cells is different in the 4 panels. This could be presented more evenly.

5. Figure 3: The validation of surface markers may be also demonstrated in freshly isolated OCs. I wonder, why CD34 was not expressed in 100%.

6. Figure 5: These pictures are partly not suitable to decide about the morphology of the differentiated cells. E.g. A-B) From my experience, hepatocytes in cell culture look different and develope the typical hexagonal shape. The bars in C and D are 200µm – is this reliable ? 

Author Response

Dear reviewer,

We are grateful for your positive comments and the constructive feedback provided. We have addressed the points raised as follows:

Comments 1. Efficiency of both methods should be compared. The authors should present absolute numbers of OCs isolated from livers of healthy and DDC treated mice.

Response 1: We agree that a direct comparison of the efficiency of the two methods is essential. In the revised manuscript, we have provided the absolute numbers of OCs isolated from healthy and DDC-treated mice. These values are presented in Results:

¨The average total number of oval cells extracted from mouse liver was 1 million in HL-OC and 6,5 million cells in DDC-OC. The extraction efficiency (number of oval cells/total number of cells extracted) for HL-OC was 3.9% while in DDC-OC it was between 10-15%, which is in agreement with the literature (29). Cell viability in both protocols was around 70%.

Comments 2. line 42: canal of hering

Response 2: The typo has been corrected in line 42. Thank you for bringing this to our attention.

Comments 3. Figure 1: Oval cells have a diameter of ~10µm (https://doi.org/10.1073/pnas.1734199100). The authors should check the micrographs and size bars as cells appear much bigger, especially in B, C, and D. If this is an artefact of the cell culture condition, it should be mentioned and discussed appropriately.

Comments 4. Figure 1: staining and appearance of the cells is different in the 4 panels. This could be presented more evenly.

Response 3 and 4: We have replaced figure 1 in an attempt to answer all reviewers, as we cannot unify the fluorescence intensity of the previous photograph. We have considered it better to include a figure at lower magnification where the cell density and morphology in culture can be observed. We thank you very much for your comments which undoubtedly improve the quality of the manuscript.

Comments 5. Figure 3: The validation of surface markers may be also demonstrated in freshly isolated OCs. I wonder, why CD34 was not expressed in 100%.

Response 5: In the bibliography it is pointed out that it is not usually a marker of adult liver but of fetal liver, although it is expressed to a greater or lesser extent by oval cells,  We believe that as the mouse grows this marker loses expression in the oval cells as it occurs in all the lines we have used. We have added this to the discussion ¨we found no significant differences in the expression of liver and surface markers between HL-OCs and DDC-OCs. We found that in both, CD34 expression was relatively low, probably because it is usually expressed in foetal liver and its expression varies and decreases in adult liver.¨, we find it very interesting and we appreciate this question.

Petersen BE, Grossbard B, Hatch H, Pi L, Deng J, Scott EW. Mouse A6-positive hepatic oval cells also express several hematopoietic stem cell markers. Hepatology. 2003 Mar; 37(3):632-40. doi: 10.1053/jhep.2003.50104. PMID: 12601361.

Comments 6. Figure 5: These pictures are partly not suitable to decide about the morphology of the differentiated cells. E.g. A-B) From my experience, hepatocytes in cell culture look different and develope the typical hexagonal shape. The bars in C and D are 200µm – is this reliable? 

Response 6: We agree with you that the typical morphology of the hepatocyte is polygonal associated to histological sections, so we have put this observation in the discussion:

¨On the other hand, we have observed that after differentiation to hepatocyte, the morphology found is not the typical polygonal hepatocyte morphology associated to histological sections, despite this, the hepatic differentiation genes were positive¨

The use of the scale in figures C and D was correct, since the cholangiocyte-like structure can only be observed complete at 10X objective, thank you for your suggestion.

We hope these revisions address the concerns raised and improve the clarity and quality of the manuscript. Thank you again for your time and valuable feedback.

Best regards,
Mariano Garcia Arranz and Rocío Olivera Salazar

Round 2

Reviewer 1 Report

Comments and Suggestions for Authors

The authors responded to the majority of the questions posed by the reviewer. The manuscript is now improved